# CatBoost: unbiased boosting with categorical features

**Liudmila Prokhorenkova**[1,2]**, Gleb Gusev**[1,2]**, Aleksandr Vorobev**[1]**,**
**Anna Veronika Dorogush**[1]**, Andrey Gulin**[1]
[1]Yandex, Moscow, Russia
[2]Moscow Institute of Physics and Technology, Dolgoprudny, Russia
{ostroumova-la, gleb57, alvor88, annaveronika, gulin}@yandex-team.ru

## Abstract

This paper presents the key algorithmic techniques behind CatBoost, a new gradient boosting toolkit. Their combination leads to CatBoost outperforming other publicly available boosting implementations in terms of quality on a variety of datasets. Two critical algorithmic advances introduced in CatBoost are the implementation of *ordered boosting*, a permutation-driven alternative to the classic algorithm, and an innovative algorithm for processing categorical features. Both techniques were created to fight a *prediction shift* caused by a special kind of target leakage present in all currently existing implementations of gradient boosting algorithms. In this paper, we provide a detailed analysis of this problem and demonstrate that proposed algorithms solve it effectively, leading to excellent empirical results.

## 1 Introduction

Gradient boosting is a powerful machine-learning technique that achieves state-of-the-art results in a variety of practical tasks. For many years, it has remained the primary method for learning problems with heterogeneous features, noisy data, and complex dependencies: web search, recommendation systems, weather forecasting, and many others [5, 26, 29, 32]. Gradient boosting is essentially a process of constructing an ensemble predictor by performing gradient descent in a functional space. It is backed by solid theoretical results that explain how strong predictors can be built by iteratively combining weaker models (*base predictors*) in a greedy manner [17].

We show in this paper that all existing implementations of gradient boosting face the following statistical issue. A prediction model $F$ obtained after several steps of boosting relies on the targets of all training examples. We demonstrate that this actually leads to a shift of the distribution of $F(\mathbf{x}_k) \mid \mathbf{x}_k$ for a training example $\mathbf{x}_k$ from the distribution of $F(\mathbf{x}) \mid \mathbf{x}$ for a test example $\mathbf{x}$. This finally leads to a *prediction shift* of the learned model. We identify this problem as a special kind of target leakage in Section 4. Further, there is a similar issue in standard algorithms of preprocessing categorical features. One of the most effective ways [6, 25] to use them in gradient boosting is converting categories to their target statistics. A target statistic is a simple statistical model itself, and it can also cause target leakage and a prediction shift. We analyze this in Section 3.

In this paper, we propose *ordering principle* to solve both problems. Relying on it, we derive *ordered boosting*, a modification of standard gradient boosting algorithm, which avoids target leakage (Section 4), and a new algorithm for processing categorical features (Section 3). Their combination is implemented as an open-source library[1] called CatBoost (for "Categorical Boosting"), which outperforms the existing state-of-the-art implementations of gradient boosted decision trees — XGBoost [8] and LightGBM [16] — on a diverse set of popular machine learning tasks (see Section 6).

## 2 Background

Assume we observe a dataset of examples $\mathcal{D} = \{(\mathbf{x}_k, y_k)\}_{k=1..n}$, where $\mathbf{x}_k = (x_k^1, \dots, x_k^m)$ is a random vector of $m$ *features* and $y_k \in \mathbb{R}$ is a *target*, which can be either binary or a numerical response. Examples $(\mathbf{x}_k, y_k)$ are independent and identically distributed according to some unknown distribution $P(\cdot, \cdot)$. The goal of a learning task is to train a function $F \colon \mathbb{R}^m \to \mathbb{R}$ which minimizes the expected loss $\mathcal{L}(F) := \mathbb{E}L(y, F(\mathbf{x}))$. Here $L(\cdot, \cdot)$ is a smooth loss function and $(\mathbf{x}, y)$ is a *test example* sampled from $P$ independently of the training set $\mathcal{D}$.

A gradient boosting procedure [12] builds iteratively a sequence of *approximations* $F^t \colon \mathbb{R}^m \to \mathbb{R}$, $t = 0, 1, \dots$ in a greedy fashion. Namely, $F^t$ is obtained from the previous approximation $F^{t-1}$ in an additive manner: $F^t = F^{t-1} + \alpha h^t$, where $\alpha$ is a *step size* and function $h^t \colon \mathbb{R}^m \to \mathbb{R}$ (a *base predictor*) is chosen from a family of functions $H$ in order to minimize the expected loss:

$$h^t = \arg\min_{h \in H} \mathcal{L}(F^{t-1} + h) = \arg\min_{h \in H} \mathbb{E}L(y, F^{t-1}(\mathbf{x}) + h(\mathbf{x})). \qquad (1)$$

The minimization problem is usually approached by the *Newton method* using a second–order approximation of $\mathcal{L}(F^{t-1} + h^t)$ at $F^{t-1}$ or by taking a *(negative) gradient step*. Both methods are kinds of functional gradient descent [10, 24]. In particular, the gradient step $h^t$ is chosen in such a way that $h^t(\mathbf{x})$ approximates $-g^t(\mathbf{x}, y)$, where $g^t(\mathbf{x}, y) := \frac{\partial L(y, s)}{\partial s}\big|_{s = F^{t-1}(\mathbf{x})}$. Usually, the least-squares approximation is used:

$$h^t = \arg\min_{h \in H} \mathbb{E}\left(-g^t(\mathbf{x}, y) - h(\mathbf{x})\right)^2. \qquad (2)$$

CatBoost is an implementation of gradient boosting, which uses binary decision trees as base predictors. A *decision tree* [4, 10, 27] is a model built by a recursive partition of the feature space $\mathbb{R}^m$ into several disjoint regions (tree nodes) according to the values of some *splitting attributes* $a$. Attributes are usually binary variables that identify that some feature $x^k$ exceeds some *threshold* $t$, that is, $a = \mathbb{1}_{\{x^k > t\}}$, where $x^k$ is either numerical or binary feature, in the latter case $t = 0.5$.[2] Each final region (leaf of the tree) is assigned to a value, which is an estimate of the response $y$ in the region for the regression task or the predicted class label in the case of classification problem.[3] In this way, a decision tree $h$ can be written as

$$h(\mathbf{x}) = \sum_{j=1}^{J} b_j \mathbb{1}_{\{\mathbf{x} \in R_j\}}, \qquad (3)$$

where $R_j$ are the disjoint regions corresponding to the leaves of the tree.

## 3 Categorical features

### 3.1 Related work on categorical features

A categorical feature is one with a discrete set of values called *categories* that are not comparable to each other. One popular technique for dealing with categorical features in boosted trees is *one-hot encoding* [7, 25], i.e., for each category, adding a new binary feature indicating it. However, in the case of high cardinality features (like, e.g., "user ID" feature), such technique leads to infeasibly large number of new features. To address this issue, one can group categories into a limited number of clusters and then apply one-hot encoding. A popular method is to group categories by *target statistics* (TS) that estimate expected target value in each category. Micci-Barreca [25] proposed to consider TS as a new numerical feature instead. Importantly, among all possible partitions of

categories into two sets, an optimal split on the training data in terms of logloss, Gini index, MSE can be found among thresholds for the numerical TS feature [4, Section 4.2.2] [11, Section 9.2.4]. In LightGBM [20], categorical features are converted to gradient statistics at each step of gradient boosting. Though providing important information for building a tree, this approach can dramatically increase (i) computation time, since it calculates statistics for each categorical value at each step, and (ii) memory consumption to store which category belongs to which node for each split based on a categorical feature. To overcome this issue, LightGBM groups tail categories into one cluster [21] and thus looses part of information. Besides, the authors claim that it is still better to convert categorical features with high cardinality to numerical features [19]. Note that TS features require calculating and storing only one number per one category.

Thus, using TS as new numerical features seems to be the most efficient method of handling categorical features with minimum information loss. TS are widely-used, e.g., in the click prediction task (click-through rates) [1, 15, 18, 22], where such categorical features as user, region, ad, publisher play a crucial role. We further focus on ways to calculate TS and leave one-hot encoding and gradient statistics out of the scope of the current paper. At the same time, we believe that the ordering principle proposed in this paper is also effective for gradient statistics.

### 3.2 Target statistics

As discussed in Section 3.1, an effective and efficient way to deal with a categorical feature $i$ is to substitute the category $x_k^i$ of $k$-th training example with *one* numeric feature equal to some *target statistic* (TS) $\hat{x}_k^i$. Commonly, it estimates the expected target $y$ conditioned by the category: $\hat{x}_k^i \approx \mathbb{E}(y \mid x^i = x_k^i)$.

**Greedy TS**  A straightforward approach is to estimate $\mathbb{E}(y \mid x^i = x_k^i)$ as the average value of $y$ over the training examples with the same category $x_k^i$ [25]. This estimate is noisy for low-frequency categories, and one usually smoothes it by some prior $p$:

$$\hat{x}_k^i = \frac{\sum_{j=1}^n \mathbb{1}_{\{x_j^i = x_k^i\}} \cdot y_j + a\,p}{\sum_{j=1}^n \mathbb{1}_{\{x_j^i = x_k^i\}} + a}, \tag{4}$$

where $a > 0$ is a parameter. A common setting for $p$ is the average target value in the dataset [25].

The problem of such *greedy* approach is *target leakage*: feature $\hat{x}_k^i$ is computed using $y_k$, the target of $\mathbf{x}_k$. This leads to a conditional shift [30]: the distribution of $\hat{x}^i \mid y$ differs for training and test examples. The following extreme example illustrates how dramatically this may affect the generalization error of the learned model. Assume $i$-th feature is categorical, all its values are unique, and for each category $A$, we have $\mathrm{P}(y = 1 \mid x^i = A) = 0.5$ for a classification task. Then, in the training dataset, $\hat{x}_k^i = \frac{y_k + ap}{1 + a}$, so it is sufficient to make only one split with threshold $t = \frac{0.5 + ap}{1 + a}$ to perfectly classify all training examples. However, for all test examples, the value of the greedy TS is $p$, and the obtained model predicts 0 for all of them if $p < t$ and predicts 1 otherwise, thus having accuracy 0.5 in both cases. To this end, we formulate the following desired property for TS:

> P1  $\mathbb{E}(\hat{x}^i \mid y = v) = \mathbb{E}(\hat{x}_k^i \mid y_k = v)$, *where* $(\mathbf{x}_k, y_k)$ *is the $k$-th training example.*

In our example above, $\mathbb{E}(\hat{x}_k^i \mid y_k) = \frac{y_k + ap}{1 + a}$ and $\mathbb{E}(\hat{x}^i \mid y) = p$ are different.

There are several ways to avoid this conditional shift. Their general idea is to compute the TS for $\mathbf{x}_k$ on a subset of examples $\mathcal{D}_k \subset \mathcal{D} \setminus \{\mathbf{x}_k\}$ excluding $\mathbf{x}_k$:

$$\hat{x}_k^i = \frac{\sum_{\mathbf{x}_j \in \mathcal{D}_k} \mathbb{1}_{\{x_j^i = x_k^i\}} \cdot y_j + a\,p}{\sum_{\mathbf{x}_j \in \mathcal{D}_k} \mathbb{1}_{\{x_j^i = x_k^i\}} + a}. \tag{5}$$

**Holdout TS**  One way is to partition the training dataset into two parts $\mathcal{D} = \hat{\mathcal{D}}_0 \sqcup \hat{\mathcal{D}}_1$ and use $\mathcal{D}_k = \hat{\mathcal{D}}_0$ for calculating the TS according to (5) and $\hat{\mathcal{D}}_1$ for training (e.g., applied in [8] for Criteo dataset). Though such *holdout* TS satisfies P1, this approach significantly reduces the amount of data used both for training the model and calculating the TS. So, it violates the following desired property:

> P2 *Effective usage of all training data for calculating TS features and for learning a model.*

**Leave-one-out TS**   At first glance, a *leave-one-out* technique might work well: take $\mathcal{D}_k = \mathcal{D} \setminus \mathbf{x}_k$ for training examples $\mathbf{x}_k$ and $\mathcal{D}_k = \mathcal{D}$ for test ones [31]. Surprisingly, it does not prevent target leakage. Indeed, consider a constant categorical feature: $x_k^i = A$ for all examples. Let $n^+$ be the number of examples with $y = 1$, then $\hat{x}_k^i = \frac{n^+ - y_k + a\,p}{n - 1 + a}$ and one can perfectly classify the training dataset by making a split with threshold $t = \frac{n^+ - 0.5 + a\,p}{n - 1 + a}$.

**Ordered TS**   CatBoost uses a more effective strategy. It relies on the ordering principle, the central idea of the paper, and is inspired by online learning algorithms which get training examples sequentially in time [1, 15, 18, 22]). Clearly, the values of TS for each example rely only on the observed history. To adapt this idea to standard offline setting, we introduce an artificial "time", i.e., a random permutation $\sigma$ of the training examples. Then, for each example, we use all the available "history" to compute its TS, i.e., take $\mathcal{D}_k = \{\mathbf{x}_j : \sigma(j) < \sigma(k)\}$ in Equation (5) for a training example and $\mathcal{D}_k = \mathcal{D}$ for a test one. The obtained *ordered* TS satisfies the requirement P1 and allows to use all training data for learning the model (P2). Note that, if we use only one random permutation, then preceding examples have TS with much higher variance than subsequent ones. To this end, CatBoost uses different permutations for different steps of gradient boosting, see details in Section 5.

# 4   Prediction shift and ordered boosting

## 4.1   Prediction shift

In this section, we reveal the problem of prediction shift in gradient boosting, which was neither recognized nor previously addressed. Like in case of TS, prediction shift is caused by a special kind of target leakage. Our solution is called *ordered boosting* and resembles the ordered TS method.

Let us go back to the gradient boosting procedure described in Section 2. In practice, the expectation in (2) is unknown and is usually approximated using the same dataset $\mathcal{D}$:

$$h^t = \arg\min_{h \in H} \frac{1}{n} \sum_{k=1}^{n} \left( -g^t(\mathbf{x}_k, y_k) - h(\mathbf{x}_k) \right)^2 . \tag{6}$$

Now we describe and analyze the following chain of shifts:

1. the conditional distribution of the gradient $g^t(\mathbf{x}_k, y_k) \mid \mathbf{x}_k$ (accounting for randomness of $\mathcal{D} \setminus \{\mathbf{x}_k\}$) is shifted from that distribution on a test example $g^t(\mathbf{x}, y) \mid \mathbf{x}$;
2. in turn, base predictor $h^t$ defined by Equation (6) is biased from the solution of Equation (2);
3. this, finally, affects the generalization ability of the trained model $F^t$.

As in the case of TS, these problems are caused by the target leakage. Indeed, gradients used at each step are estimated using the target values of the same data points the current model $F^{t-1}$ was built on. However, the conditional distribution $F^{t-1}(\mathbf{x}_k) \mid \mathbf{x}_k$ for a training example $\mathbf{x}_k$ is shifted, in general, from the distribution $F^{t-1}(\mathbf{x}) \mid \mathbf{x}$ for a test example $\mathbf{x}$. We call this a *prediction shift*.

**Related work on prediction shift**   The shift of gradient conditional distribution $g^t(\mathbf{x}_k, y_k) \mid \mathbf{x}_k$ was previously mentioned in papers on boosting [3, 13] but was not formally defined. Moreover, even the existence of non-zero shift was not proved theoretically. Based on the out-of-bag estimation [2], Breiman proposed *iterated bagging* [3] which constructs a bagged weak learner at each iteration on the basis of "out-of-bag" residual estimates. However, as we formally show in Section E of the supplementary material, such residual estimates are still shifted. Besides, the bagging scheme increases learning time by factor of the number of data buckets. Subsampling of the dataset at each iteration proposed by Friedman [13] addresses the problem much more heuristically and also only alleviates it.

**Analysis of prediction shift**   We formally analyze the problem of prediction shift in a simple case of a regression task with the quadratic loss function $L(y, \hat{y}) = (y - \hat{y})^2$.[4] In this case, the negative gradient $-g^t(\mathbf{x}_k, y_k)$ in Equation (6) can be substituted by the residual function $r^{t-1}(\mathbf{x}_k, y_k) := y_k - F^{t-1}(\mathbf{x}_k)$.[5] Assume we have $m = 2$ features $x^1, x^2$ that are i.i.d. Bernoulli random variables

with $p = 1/2$ and $y = f^*(\mathbf{x}) = c_1 x^1 + c_2 x^2$. Assume we make $N = 2$ steps of gradient boosting with decision stumps (trees of depth 1) and step size $\alpha = 1$. We obtain a model $F = F^2 = h^1 + h^2$. W.l.o.g., we assume that $h^1$ is based on $x^1$ and $h^2$ is based on $x^2$, what is typical for $|c_1| > |c_2|$ (here we set some asymmetry between $x^1$ and $x^2$).

**Theorem 1** *1. If two independent samples $\mathcal{D}_1$ and $\mathcal{D}_2$ of size $n$ are used to estimate $h^1$ and $h^2$, respectively, using Equation* (6)*, then $\mathbb{E}_{\mathcal{D}_1, \mathcal{D}_2} F^2(\mathbf{x}) = f^*(\mathbf{x}) + O(1/2^n)$ for any $\mathbf{x} \in \{0, 1\}^2$.*
*2. If the same dataset $\mathcal{D} = \mathcal{D}_1 = \mathcal{D}_2$ is used in Equation* (6) *for both $h^1$ and $h^2$, then $\mathbb{E}_{\mathcal{D}} F^2(\mathbf{x}) = f^*(\mathbf{x}) - \frac{1}{n-1} c_2 (x^2 - \frac{1}{2}) + O(1/2^n)$.*

This theorem means that the trained model is an unbiased estimate of the true dependence $y = f^*(\mathbf{x})$, when we use independent datasets at each gradient step.[6] On the other hand, if we use the same dataset at each step, we suffer from a bias $-\frac{1}{n-1} c_2 (x^2 - \frac{1}{2})$, which is inversely proportional to the data size $n$. Also, the value of the bias can depend on the relation $f^*$: in our example, it is proportional to $c_2$. We track the chain of shifts for the second part of Theorem 1 in a sketch of the proof below, while the full proof of Theorem 1 is available in the supplementary material (Section A).

*Sketch of the proof.* Denote by $\xi_{st}$, $s, t \in \{0, 1\}$, the number of examples $(\mathbf{x}_k, y_k) \in \mathcal{D}$ with $\mathbf{x}_k = (s, t)$. We have $h^1(s, t) = c_1 s + \frac{c_2 \xi_{s1}}{\xi_{s0} + \xi_{s1}}$. Its expectation $\mathbb{E}(h^1(\mathbf{x}))$ on a test example $\mathbf{x}$ equals $c_1 x^1 + \frac{c_2}{2}$. At the same time, the expectation $\mathbb{E}(h^1(\mathbf{x}_k))$ on a training example $\mathbf{x}_k$ is different and equals $(c_1 x^1 + \frac{c_2}{2}) - c_2 (\frac{2 x^2 - 1}{n}) + O(2^{-n})$. That is, we experience a prediction shift of $h^1$. As a consequence, the expected value of $h^2(\mathbf{x})$ is $\mathbb{E}(h^2(\mathbf{x})) = c_2 (x^2 - \frac{1}{2})(1 - \frac{1}{n-1}) + O(2^{-n})$ on a test example $\mathbf{x}$ and $\mathbb{E}(h^1(\mathbf{x}) + h^2(\mathbf{x})) = f^*(\mathbf{x}) - \frac{1}{n-1} c_2 (x^2 - \frac{1}{2}) + O(1/2^n)$. $\square$

Finally, recall that greedy TS $\hat{x}^i$ can be considered as a simple statistical model predicting the target $y$ and it suffers from a similar problem, conditional shift of $\hat{x}^i_k \mid y_k$, caused by the target leakage, i.e., using $y_k$ to compute $\hat{x}^i_k$.

## 4.2 Ordered boosting

Here we propose a boosting algorithm which does not suffer from the prediction shift problem described in Section 4.1. Assuming access to an unlimited amount of training data, we can easily construct such an algorithm. At each step of boosting, we sample a new dataset $\mathcal{D}_t$ independently and obtain unshifted residuals by applying the current model to new training examples. In practice, however, labeled data is limited. Assume that we learn a model with $I$ trees. To make the residual $r^{I-1}(\mathbf{x}_k, y_k)$ unshifted, we need to have $F^{I-1}$ trained without the example $\mathbf{x}_k$. Since we need unbiased residuals for all training examples, no examples may be used for training $F^{I-1}$, which at first glance makes the training process impossible. However, it is possible to maintain a set of models differing by examples used for their training. Then, for calculating the residual on an example, we use a model trained without it. In order to construct such a set of models, we can use the ordering principle previously applied to TS in Section 3.2. To illustrate the idea, assume that we take one random permutation $\sigma$ of the training examples and maintain $n$ different *supporting* models $M_1, \ldots, M_n$ such that the model $M_i$ is learned using only the first $i$ examples in the permutation. At each step, in order to obtain the residual for $j$-th sample, we use the model $M_{j-1}$ (see Figure 1). The resulting Algorithm 1 is called *ordered boosting* below. Unfortunately, this algorithm is not feasible in most practical tasks due to the need of training $n$ different models, what increase the complexity and memory requirements by $n$ times. In CatBoost, we implemented a modification of this algorithm on the basis of the gradient boosting algorithm with decision trees as base predictors (GBDT) described in Section 5.

**Ordered boosting with categorical features** In Sections 3.2 and 4.2 we proposed to use random permutations $\sigma_{cat}$ and $\sigma_{boost}$ of training examples for the TS calculation and for ordered boosting, respectively. Combining them in one algorithm, we should take $\sigma_{cat} = \sigma_{boost}$ to avoid prediction shift. This guarantees that target $y_i$ is not used for training $M_i$ (neither for the TS calculation, nor for the gradient estimation). See Section F of the supplementary material for theoretical guarantees. Empirical results confirming the importance of having $\sigma_{cat} = \sigma_{boost}$ are presented in Section G of the supplementary material.

# 5 Practical implementation of ordered boosting

CatBoost has two boosting modes, *Ordered* and *Plain*. The latter mode is the standard GBDT algorithm with inbuilt ordered TS. The former mode presents an efficient modification of Algorithm 1. A formal description of the algorithm is included in Section B of the supplementary material. In this section, we overview the most important implementation details.

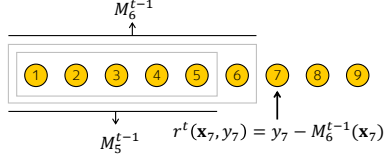

Figure 1: Ordered boosting principle, examples are ordered according to $\sigma$.

---

**Algorithm 1:** Ordered boosting

**input** : $\{(\mathbf{x}_k, y_k)\}_{k=1}^n$, $I$;

$\sigma \leftarrow$ random permutation of $[1, n]$ ;
$M_i \leftarrow 0$ for $i = 1..n$;
**for** $t \leftarrow 1$ **to** $I$ **do**
    **for** $i \leftarrow 1$ **to** $n$ **do**
        $r_i \leftarrow y_i - M_{\sigma(i)-1}(\mathbf{x}_i)$;
    **for** $i \leftarrow 1$ **to** $n$ **do**
        $\Delta M \leftarrow$
        $LearnModel((\mathbf{x}_j, r_j) :$
        $\sigma(j) \leq i)$;
        $M_i \leftarrow M_i + \Delta M$ ;

**return** $M_n$

---

**Algorithm 2:** Building a tree in CatBoost

**input** : $M$, $\{(\mathbf{x}_i, y_i)\}_{i=1}^n$, $\alpha$, $L$, $\{\sigma_i\}_{i=1}^s$, $Mode$

$grad \leftarrow CalcGradient(L, M, y)$;
$r \leftarrow random(1, s)$;
**if** $Mode = Plain$ **then**
    $G \leftarrow (grad_r(i)$ for $i = 1..n)$;
**if** $Mode = Ordered$ **then**
    $G \leftarrow (grad_{r,\sigma_r(i)-1}(i)$ for $i = 1..n)$;
$T \leftarrow$ empty tree;
**foreach** *step of top-down procedure* **do**
    **foreach** *candidate split c* **do**
        $T_c \leftarrow$ add split $c$ to $T$;
        **if** $Mode = Plain$ **then**
            $\Delta(i) \leftarrow \text{avg}(grad_r(p)$ for
            $p : leaf_r(p) = leaf_r(i))$ for $i = 1..n$;
        **if** $Mode = Ordered$ **then**
            $\Delta(i) \leftarrow \text{avg}(grad_{r,\sigma_r(i)-1}(p)$ for
            $p : leaf_r(p) = leaf_r(i), \sigma_r(p) < \sigma_r(i))$
            for $i = 1..n$;
        $loss(T_c) \leftarrow \cos(\Delta, G)$
    $T \leftarrow \arg\min_{T_c}(loss(T_c))$
**if** $Mode = Plain$ **then**
    $M_{r'}(i) \leftarrow M_{r'}(i) - \alpha \, \text{avg}(grad_{r'}(p)$ for
    $p : leaf_{r'}(p) = leaf_{r'}(i))$ for $r' = 1..s, i = 1..n$;
**if** $Mode = Ordered$ **then**
    $M_{r',j}(i) \leftarrow M_{r',j}(i) - \alpha \, \text{avg}(grad_{r',j}(p)$ for
    $p : leaf_{r'}(p) = leaf_{r'}(i), \sigma_{r'}(p) \leq j)$ for $r' = 1..s$,
    $i = 1..n, j \geq \sigma_{r'}(i) - 1$;
**return** $T, M$

---

At the start, CatBoost generates $s + 1$ independent random permutations of the training dataset. The permutations $\sigma_1, \ldots, \sigma_s$ are used for evaluation of splits that define tree structures (i.e., the internal nodes), while $\sigma_0$ serves for choosing the leaf values $b_j$ of the obtained trees (see Equation (3)). For examples with short history in a given permutation, both TS and predictions used by ordered boosting ($M_{\sigma(i)-1}(\mathbf{x}_i)$ in Algorithm 1) have a high variance. Therefore, using only one permutation may increase the variance of the final model predictions, while several permutations allow us to reduce this effect in a way we further describe. The advantage of several permutations is confirmed by our experiments in Section 6.

**Building a tree**  In CatBoost, base predictors are oblivious decision trees [9, 14] also called decision tables [23]. Term oblivious means that the same splitting criterion is used across an entire level of the tree. Such trees are balanced, less prone to overfitting, and allow speeding up execution at testing time significantly. The procedure of building a tree in CatBoost is described in Algorithm 2.

In the Ordered boosting mode, during the learning process, we maintain the supporting models $M_{r,j}$, where $M_{r,j}(i)$ is the current prediction for the $i$-th example based on the first $j$ examples in the permutation $\sigma_r$. At each iteration $t$ of the algorithm, we sample a random permutation $\sigma_r$ from $\{\sigma_1, \ldots, \sigma_s\}$ and construct a tree $T_t$ on the basis of it. First, for categorical features, all TS are computed according to this permutation. Second, the permutation affects the tree learning procedure.

Table 1: Computational complexity.

| Procedure | CalcGradient | Build $T$ | Calc all $b_j^t$ | Update $M$ | Calc ordered TS |
|---|---|---|---|---|---|
| Complexity for iteration $t$ | $O(s \cdot n)$ | $O(|C| \cdot n)$ | $O(n)$ | $O(s \cdot n)$ | $O(N_{TS,t} \cdot n)$ |

Namely, based on $M_{r,j}(i)$, we compute the corresponding gradients $grad_{r,j}(i) = \frac{\partial L(y_i,s)}{\partial s}\big|_{s=M_{r,j}(i)}$. Then, while constructing a tree, we approximate the gradient $G$ in terms of the cosine similarity $\cos(\cdot, \cdot)$, where, for each example $i$, we take the gradient $grad_{r,\sigma(i)-1}(i)$ (it is based only on the previous examples in $\sigma_r$). At the candidate splits evaluation step, the leaf value $\Delta(i)$ for example $i$ is obtained individually by averaging the gradients $grad_{r,\sigma_r(i)-1}$ of the preceding examples $p$ lying in the same leaf $leaf_r(i)$ the example $i$ belongs to. Note that $leaf_r(i)$ depends on the chosen permutation $\sigma_r$, because $\sigma_r$ can influence the values of ordered TS for example $i$. When the tree structure $T_t$ (i.e., the sequence of splitting attributes) is built, we use it to boost all the models $M_{r',j}$. Let us stress that *one common* tree structure $T_t$ is used for all the models, but this tree is added to different $M_{r',j}$ with different sets of leaf values depending on $r'$ and $j$, as described in Algorithm 2.

The Plain boosting mode works similarly to a standard GBDT procedure, but, if categorical features are present, it maintains $s$ supporting models $M_r$ corresponding to TS based on $\sigma_1, \ldots, \sigma_s$.

**Choosing leaf values**  Given all the trees constructed, the leaf values of the final model $F$ are calculated by the standard gradient boosting procedure equally for both modes. Training examples $i$ are matched to leaves $leaf_0(i)$, i.e., we use permutation $\sigma_0$ to calculate TS here. When the final model $F$ is applied to a new example at testing time, we use TS calculated on the whole training data according to Section 3.2.

**Complexity**  In our practical implementation, we use one important trick, which significantly reduces the computational complexity of the algorithm. Namely, in the Ordered mode, instead of $O(s\,n^2)$ values $M_{r,j}(i)$, we store and update only the values $M'_{r,j}(i) := M_{r,2^j}(i)$ for $j = 1, \ldots, \lceil \log_2 n \rceil$ and all $i$ with $\sigma_r(i) \leq 2^{j+1}$, what reduces the number of maintained supporting predictions to $O(s\,n)$. See Section B of the supplementary material for the pseudocode of this modification of Algorithm 2.

In Table 1, we present the computational complexity of different components of both CatBoost modes per one iteration (see Section C.1 of the supplementary material for the proof). Here $N_{TS,t}$ is the number of TS to be calculated at the iteration $t$ and $C$ is the set of candidate splits to be considered at the given iteration. It follows that our implementation of ordered boosting with decision trees has the same asymptotic complexity as the standard GBDT with ordered TS. In comparison with other types of TS (Section 3.2), ordered TS slow down by $s$ times the procedures $CalcGradient$, updating supporting models $M$, and computation of TS.

**Feature combinations**  Another important detail of CatBoost is using combinations of categorical features as additional categorical features which capture high-order dependencies like joint information of user ID and ad topic in the task of ad click prediction. The number of possible combinations grows exponentially with the number of categorical features in the dataset, and it is infeasible to process all of them. CatBoost constructs combinations in a greedy way. Namely, for each split of a tree, CatBoost combines (concatenates) all categorical features (and their combinations) already used for previous splits in the current tree with all categorical features in the dataset. Combinations are converted to TS on the fly.

**Other important details**  Finally, let us discuss two options of the CatBoost algorithm not covered above. The first one is subsampling of the dataset at each iteration of boosting procedure, as proposed by Friedman [13]. We claimed earlier in Section 4.1 that this approach alone cannot fully avoid the problem of prediction shift. However, since it has proved effective, we implemented it in both modes of CatBoost as a Bayesian bootstrap procedure. Specifically, before training a tree according to Algorithm 2, we assign a weight $w_i = a_i^t$ to each example $i$, where $a_i^t$ are generated according to the Bayesian bootstrap procedure (see [28, Section 2]). These weights are used as multipliers for gradients $grad_r(i)$ and $grad_{r,j}(i)$, when we calculate $\Delta(i)$ and the components of the vector $\Delta - G$ to define $loss(T_c)$.

The second option deals with first several examples in a permutation. For examples $i$ with small values $\sigma_r(i)$, the variance of $grad_{r,\sigma_r(i)-1}(i)$ can be high. Therefore, we discard $\Delta(i)$ from the beginning of the permutation, when we calculate $loss(T_c)$ in Algorithm 2. Particularly, we eliminate the corresponding components of vectors $G$ and $\Delta$ when calculating the cosine similarity between them.

## 6    Experiments

**Comparison with baselines**    We compare our algorithm with the most popular open-source libraries — XGBoost and LightGBM — on several well-known machine learning tasks. The detailed description of the experimental setup together with dataset descriptions is available in the supplementary material (Section D). The source code of the experiment is available, and the results can be reproduced.[7] For all learning algorithms, we preprocess categorical features using the ordered TS method described in Section 3.2. The parameter tuning and training were performed on 4/5 of the data and the testing was performed on the remaining 1/5.[8] The results measured by logloss and zero-one loss are presented in Table 2 (the absolute values for the baselines can be found in Section G of the supplementary material). For CatBoost, we used Ordered boosting mode in this experiment.[9] One can see that CatBoost outperforms other algorithms on all the considered datasets. We also measured statistical significance of improvements presented in Table 2: except three datasets (Appetency, Churn and Upselling) the improvements are statistically significant with p-value $\ll 0.01$ measured by the paired one-tailed t-test.

To demonstrate that our implementation of plain boosting is an appropriate baseline for our research, we show that a *raw setting* of CatBoost provides state-of-the-art quality. Particularly, we take a setting of CatBoost, which is close to classical GBDT [12], and compare it with the baseline boosting implementations in Section G of the supplementary material. Experiments show that this raw setting differs from the baselines insignificantly.

Table 2: Comparison with baselines: logloss / zero-one loss (relative increase for baselines).

| | CatBoost | LightGBM | XGBoost |
|---|---|---|---|
| Adult | **0.270 / 0.127** | +2.4% / +1.9% | +2.2% / +1.0% |
| Amazon | **0.139 / 0.044** | +17% / +21% | +17% / +21% |
| Click | **0.392 / 0.156** | +1.2% / +1.2% | +1.2% / +1.2% |
| Epsilon | **0.265 / 0.109** | +1.5% / +4.1% | +11% / +12% |
| Appetency | **0.072 / 0.018** | +0.4% / +0.2% | +0.4% / +0.7% |
| Churn | **0.232 / 0.072** | +0.1% / +0.6% | +0.5% / +1.6% |
| Internet | **0.209 / 0.094** | +6.8% / +8.6% | +7.9% / +8.0% |
| Upselling | **0.166 / 0.049** | +0.3% / +0.1% | +0.04% / +0.3% |
| Kick | **0.286 / 0.095** | +3.5% / +4.4% | +3.2% / +4.1% |

Table 3: Plain boosting mode: logloss, zero-one loss and their change relative to Ordered boosting mode.

| | Logloss | Zero-one loss |
|---|---|---|
| Adult | 0.272 (+1.1%) | 0.127 (-0.1%) |
| Amazon | 0.139 (-0.6%) | 0.044 (-1.5%) |
| Click | 0.392 (-0.05%) | 0.156 (+0.19%) |
| Epsilon | 0.266 (+0.6%) | 0.110 (+0.9%) |
| Appetency | 0.072 (+0.5%) | 0.018 (+1.5%) |
| Churn | 0.232 (-0.06%) | 0.072 (-0.17%) |
| Internet | 0.217 (+3.9%) | 0.099 (+5.4%) |
| Upselling | 0.166 (+0.1%) | 0.049 (+0.4%) |
| Kick | 0.285 (-0.2%) | 0.095 (-0.1%) |

We also empirically analyzed the running times of the algorithms on Epsilon dataset. The details of the comparison can be found in the supplementary material (Section C.2). To summarize, we obtained that CatBoost Plain and LightGBM are the fastest ones followed by Ordered mode, which is about 1.7 times slower.

**Ordered and Plain modes**    In this section, we compare two essential boosting modes of CatBoost: Plain and Ordered. First, we compared their performance on all the considered datasets, the results are presented in Table 3. It can be clearly seen that Ordered mode is particularly useful on small datasets. Indeed, the largest benefit from Ordered is observed on Adult and Internet datasets, which are relatively small (less than 40K training examples), which supports our hypothesis that a higher bias negatively affects the performance. Indeed, according to Theorem 1 and our reasoning in Section 4.1, bias is expected to be larger for smaller datasets (however, it can also depend on other properties of the dataset, e.g., on the dependency between features and target). In order to further

validate this hypothesis, we make the following experiment: we train CatBoost in Ordered and Plain modes on randomly filtered datasets and compare the obtained losses, see Figure 2. As we expected, for smaller datasets the relative performance of Plain mode becomes worse. To save space, here we present the results only for logloss; the figure for zero-one loss is similar.

We also compare Ordered and Plain modes in the above-mentioned raw setting of CatBoost in Section G of the supplementary material and conclude that the advantage of Ordered mode is not caused by interaction with specific CatBoost options.

Table 4: Comparison of target statistics, relative change in logloss / zero-one loss compared to ordered TS.

|  | Greedy | Holdout | Leave-one-out |
|---|---|---|---|
| Adult | +1.1% / +0.8% | +2.1% / +2.0% | +5.5% / +3.7% |
| Amazon | +40% / +32% | +8.3% / +8.3% | +4.5% / +5.6% |
| Click | +13% / +6.7% | +1.5% / +0.5% | +2.7% / +0.9% |
| Appetency | +24% / +0.7% | +1.6% / -0.5% | +8.5% / +0.7% |
| Churn | +12% / +2.1% | +0.9% / +1.3% | +1.6% / +1.8% |
| Internet | +33% / +22% | +2.6% / +1.8% | +27% / +19% |
| Upselling | +57% / +50% | +1.6% / +0.9% | +3.9% / +2.9% |
| Kick | +22% / +28% | +1.3% / +0.32% | +3.7% / +3.3% |

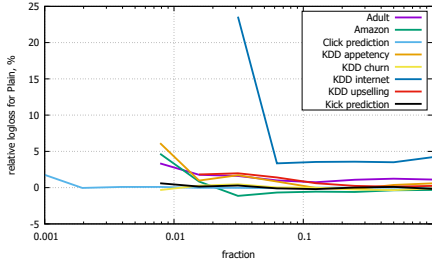

Figure 2: Relative error of Plain boosting mode compared to Ordered boosting mode depending on the fraction of the dataset.

**Analysis of target statistics**   We compare different TSs introduced in Section 3.2 as options of CatBoost in Ordered boosting mode keeping all other algorithmic details the same; the results can be found in Table 4. Here, to save space, we present only relative increase in loss functions for each algorithm compared to CatBoost with ordered TS. Note that the ordered TS used in CatBoost significantly outperform all other approaches. Also, among the baselines, the holdout TS is the best for most of the datasets since it does not suffer from conditional shift discussed in Section 3.2 (P1); still, it is worse than CatBoost due to less effective usage of training data (P2). Leave-one-out is usually better than the greedy TS, but it can be much worse on some datasets, e.g., on Adult. The reason is that the greedy TS suffer from low-frequency categories, while the leave-one-out TS suffer also from high-frequency ones, and on Adult all the features have high frequency.

Finally, let us note that in Table 4 we combine Ordered mode of CatBoost with different TSs. To generalize these results, we also made a similar experiment by combining different TS with Plain mode, used in standard gradient boosting. The obtained results and conclusions turned out to be very similar to the ones discussed above.

**Feature combinations**   The effect of feature combinations discussed in Section 5 is demonstrated in Figure 1 of the supplementary material. In average, changing the number $c_{max}$ of features allowed to be combined from 1 to 2 provides an outstanding improvement of logloss by $1.86\%$ (reaching $11.3\%$), changing from 1 to 3 yields $2.04\%$, and further increase of $c_{max}$ does not influence the performance significantly.

**Number of permutations**   The effect of the number $s$ of permutations on the performance of CatBoost is presented in Figure 2 of the supplementary material. In average, increasing $s$ slightly decreases logloss, e.g., by $0.19\%$ for $s = 3$ and by $0.38\%$ for $s = 9$ compared to $s = 1$.

# 7   Conclusion

In this paper, we identify and analyze the problem of prediction shifts present in all existing implementations of gradient boosting. We propose a general solution, ordered boosting with ordered TS, which solves the problem. This idea is implemented in CatBoost, which is a new gradient boosting library. Empirical results demonstrate that CatBoost outperforms leading GBDT packages and leads to new state-of-the-art results on common benchmarks.

**Acknowledgments**

We are very grateful to Mikhail Bilenko for important references and advices that lead to theoretical analysis of this paper, as well as suggestions on the presentation. We also thank Pavel Serdyukov for many helpful discussions and valuable links, Nikita Kazeev, Nikita Dmitriev, Stanislav Kirillov and Victor Omelyanenko for help with experiments.

## Footnotes

[1]https://github.com/catboost/catboost

[2]Alternatively, non-binary splits can be used, e.g., a region can be split according to all values of a categorical feature. However, such splits, compared to binary ones, would lead to either shallow trees (unable to capture complex dependencies) or to very complex trees with exponential number of terminal nodes (having weaker target statistics in each of them). According to [4], the tree complexity has a crucial effect on the accuracy of the model and less complex trees are less prone to overfitting.

[3]In a regression task, splitting attributes and leaf values are usually chosen by the least–squares criterion. Note that, in gradient boosting, a tree is constructed to approximate the negative gradient (see Equation (2)), so it solves a regression problem.

[4]We restrict the rest of Section 4 to this case, but the approaches of Section 4.2 are applicable to other tasks.

[5]Here we removed the multiplier 2, what does not matter for further analysis.

[6]Up to an exponentially small term, which occurs for a technical reason.

[7]`https://github.com/catboost/benchmarks/tree/master/quality_benchmarks`

[8]For Epsilon, we use default parameters instead of parameter tuning due to large running time for all algorithms. We tune only the number of trees to avoid overfitting.

[9]The numbers for CatBoost in Table 2 may slightly differ from the corresponding numbers in our GitHub repository, since we use another version of CatBoost with all the discussed features implemented.

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
