[Supplementary Material]

# Supplementary material for
# CatBoost: unbiased boosting with categorical features

**Liudmila Prokhorenkova**[1,2]**, Gleb Gusev**[1,2]**, Aleksandr Vorobev**[1]**,**
**Anna Veronika Dorogush**[1]**, Andrey Gulin**[1]
[1]Yandex, Moscow, Russia
[2]Moscow Institute of Physics and Technology, Dolgoprudny, Russia
{ostroumova-la, gleb57, alvor88, annaveronika, gulin}@yandex-team.ru

## A  Proof of Theorem 1

### A.1  Proof for the case $\mathcal{D}_1 = \mathcal{D}_2$

Let us denote by $A$ the event that each leaf in both stumps $h^1$ and $h^2$ contains at least one example, i.e., there exists at least one $\mathbf{x} \in \mathcal{D}$ with $\mathbf{x}^i = s$ for all $i \in \{1, 2\}$, $s \in \{0, 1\}$. All further reasonings are given conditioning on $A$. Note that the probability of $A$ is $1 - O\left(2^{-n}\right)$, therefore we can assign an arbitrary value to any empty leaf during the learning process, and the choice of the value will affect all expectations we calculate below by $O\left(2^{-n}\right)$.

Denote by $\xi_{st}$, $s, t \in \{0, 1\}$, the number of examples $\mathbf{x}_k \in \mathcal{D}$ with $\mathbf{x}_k = (s, t)$. The value of the first stump $h^1$ in the region $\{x^1 = s\}$ is the average value of $y_k$ over examples from $\mathcal{D}$ belonging to this region. That is,

$$h^1(0, t) = \frac{\sum_{j=1}^{n} c_2 \mathbb{1}_{\{x_j = (0,1)\}}}{\sum_{j=1}^{n} \mathbb{1}_{\{x_j^1 = 0\}}} = \frac{c_2 \xi_{01}}{\xi_{00} + \xi_{01}},$$

$$h^1(1, t) = \frac{\sum_{j=1}^{n} c_1 \mathbb{1}_{\{x_j^1 = 1\}} + c_2 \mathbb{1}_{\{x_j = (1,1)\}}}{\sum_{j=1}^{n} \mathbb{1}_{\{x_j^1 = 1\}}} = c_1 + \frac{c_2 \xi_{11}}{\xi_{10} + \xi_{11}}.$$

Summarizing, we obtain

$$h^1(s, t) = c_1 s + \frac{c_2 \xi_{s1}}{\xi_{s0} + \xi_{s1}}. \tag{1}$$

Note that, by conditioning on $A$, we guarantee that the denominator $\xi_{s0} + \xi_{s1}$ is not equal to zero.

Now we derive the expectation $\mathbb{E}(h^1(\mathbf{x}))$ of prediction $h^1$ for a test example $\mathbf{x} = (s, t)$.

It is easy to show that $\mathbb{E}\left(\frac{\xi_{s1}}{\xi_{s0} + \xi_{s1}} \mid A\right) = \frac{1}{2}$. Indeed, due to the symmetry we have $\mathbb{E}\left(\frac{\xi_{s1}}{\xi_{s0} + \xi_{s1}} \mid A\right) = \mathbb{E}\left(\frac{\xi_{s0}}{\xi_{s0} + \xi_{s1}} \mid A\right)$ and the sum of these expectations is $\mathbb{E}\left(\frac{\xi_{s0} + \xi_{s1}}{\xi_{s0} + \xi_{s1}} \mid A\right) = 1$. So, by taking the expectation of (1), we obtain the following proposition.

**Proposition 1** *We have* $\mathbb{E}(h^1(s, t) \mid A) = c_1 s + \frac{c_2}{2}$.

It means that the conditional expectation $\mathbb{E}(h^1(\mathbf{x}) \mid \mathbf{x} = (s, t), A)$ on a test example $\mathbf{x}$ equals $c_1 s + \frac{c_2}{2}$, since $\mathbf{x}$ and $h^1$ are independent.

**Prediction shift of $h^1$**  In this paragraph, we show that the conditional expectation $\mathbb{E}(h^1(\mathbf{x}_l) \mid \mathbf{x}_l = (s, t), A)$ on a training example $\mathbf{x}_l$ is shifted for any $l = 1, \ldots, n$, because the model $h^1$ is fitted to $\mathbf{x}_l$. This is an auxiliary result, which is not used directly for proving the theorem, but helps to track the chain of obtained shifts.

**Proposition 2** *The conditional expectation is*

$$\mathbb{E}(h^1(\mathbf{x}_l) \mid \mathbf{x}_l = (s,t), A) = c_1 s + \frac{c_2}{2} - c_2 \left( \frac{2t-1}{n} \right) + O(2^{-n}).$$

*Proof.* Let us introduce the following notation

$$\alpha_{sk} = \frac{\mathbb{1}_{\{x_k = (s,1)\}}}{\xi_{s0} + \xi_{s1}}.$$

Then, we can rewrite the conditional expectation as

$$c_1 s + c_2 \sum_{k=1}^{n} \mathbb{E}(\alpha_{sk} \mid \mathbf{x}_l = (s,t), A).$$

Lemma 1 below implies that $\mathbb{E}(\alpha_{sl} \mid \mathbf{x}_l = (s,t), A) = \frac{2t}{n}$. For $k \neq l$, we have

$$\mathbb{E}(\alpha_{sk} \mid \mathbf{x}_l = (s,t), A) = \frac{1}{4} \mathbb{E}\left( \frac{1}{\xi_{s0} + \xi_{s1}} \mid \mathbf{x}_l = (s,t), \mathbf{x}_k = (s,1), A \right)$$

$$= \frac{1}{2n} \left( 1 - \frac{1}{n-1} + \frac{n-2}{(2^{n-1}-2)(n-1)} \right)$$

due to Lemma 2 below. Finally, we obtain

$$\mathbb{E}(h^1(\mathbf{x}_l) \mid \mathbf{x}_l = (s,t)) = c_1 s + c_2 \left( \frac{2t}{n} + (n-1)\frac{1}{2n}\left( 1 - \frac{1}{n-1} \right) \right)$$

$$+ O\left( 2^{-n} \right) = c_1 s + \frac{c_2}{2} - c_2 \left( \frac{2t-1}{n} \right) + O(2^{-n}).$$

$\square$

**Lemma 1** $\mathbb{E}\left( \frac{1}{\xi_{s0}+\xi_{s1}} \mid \mathbf{x}_1 = (s,t), A \right) = \frac{2}{n}.$

*Proof.* Note that given $\mathbf{x}_1 = (s,t)$, $A$ corresponds to the event that there is an example with $x^1 = 1 - s$ and (possibly another) example with $x^2 = 1 - t$ among $\mathbf{x}_2, \ldots, \mathbf{x}_n$.

Note that $\xi_{s0} + \xi_{s1} = \sum_{j=1}^{n} \mathbb{1}_{\{x_j^1 = s\}}$. For $k = 1, \ldots, n-1$, we have

$$P(\xi_{s0} + \xi_{s1} = k \mid \mathbf{x}_1 = (s,t), A) = \frac{P(\xi_{s0} + \xi_{s1} = k, A \mid \mathbf{x}_1 = (s,t))}{P(A \mid \mathbf{x}_1 = (s,t))} = \frac{\binom{n-1}{k-1}}{2^{n-1}\left( 1 - 2^{-(n-1)} \right)},$$

since $\mathbb{1}_{\{x_1^1 = s\}} = 1$ when $\mathbf{x}_1 = (s,t)$ with probability 1, $\sum_{j=2}^{n} \mathbb{1}_{\{x_j^1 = s\}}$ is a binomial variable independent of $\mathbf{x}_1$, and an example with $x^1 = 1 - s$ exists whenever $\xi_{s0} + \xi_{s1} = k < n$ and $\mathbf{x}_1 = (s,t)$ (while the existence of one with $x^2 = 1 - t$ is an independent event). Therefore, we have

$$\mathbb{E}\left( \frac{1}{\xi_{s0}+\xi_{s1}} \mid \mathbf{x}_1 = (s,t), A \right) = \sum_{k=1}^{n-1} \frac{1}{k} \frac{\binom{n-1}{k-1}}{2^{n-1}-1} = \frac{1}{n(2^{n-1}-1)} \sum_{k=1}^{n-1} \binom{n}{k} = \frac{2}{n}.$$

$\square$

**Lemma 2** *We have*

$$\mathbb{E}\left( \frac{1}{\xi_{s0}+\xi_{s1}} \mid \mathbf{x}_1 = (s,t_1), \mathbf{x}_2 = (s,t_2), A \right) = \frac{2}{n} \left( 1 - \frac{1}{n-1} + \frac{n-2}{(2^{n-1}-2)(n-1)} \right).$$

*Proof.* Similarly to the previous proof, for $k = 2, \ldots, n-1$, we have

$$P\left( \xi_{s0} + \xi_{s1} = k \mid \mathbf{x}_1 = (s,t_1), \mathbf{x}_2 = (s,t_2), A \right) = \frac{\binom{n-2}{k-2}}{2^{n-2}\left( 1 - 2^{-(n-2)} \right)}.$$

Therefore,

$$\mathbb{E}\left(\frac{1}{\xi_{s0}+\xi_{s1}} \mid \mathbf{x}_1=(s,t_1), \mathbf{x}_2=(s,t_2), A\right) = \frac{1}{2^{n-2}\left(1-2^{-(n-1)}\right)} \sum_{k=2}^{n-1} \frac{\binom{n-2}{k-2}}{k}$$

$$= \frac{1}{2^{n-2}-1} \sum_{k=2}^{n-1} \binom{n-2}{k-2}\left(\frac{1}{k-1}-\frac{1}{(k-1)k}\right)$$

$$= \frac{1}{2^{n-2}-1} \sum_{k=2}^{n-1} \left(\frac{1}{n-1}\binom{n-1}{k-1}-\frac{1}{n(n-1)}\binom{n}{k}\right) =$$

$$= \frac{1}{2^{n-2}-1} \left(\frac{1}{n-1}(2^{n-1}-2)-\frac{1}{n(n-1)}(2^n-n-2)\right) =$$

$$= \frac{2}{n}\left(1-\frac{1}{n-1}+\frac{n-2}{(2^{n-1}-2)(n-1)}\right).$$

□

**Bias of the model $h^1+h^2$**   Proposition 2 shows that the values of the model $h^1$ on training examples are shifted with respect to the ones on test examples. The next step is to show how this can lead to a bias of the trained model, if we use the same dataset for building both $h^1$ and $h^2$. Namely, we derive the expected value of $h^1(s,t)+h^2(s,t)$ and obtain a bias according to the following result.

**Proposition 3** *If both $h^1$ and $h^2$ are built using the same dataset $\mathcal{D}$, then*

$$\mathbb{E}\left(h^1(s,t)+h^2(s,t) \mid A\right) = f^*(s,t) - \frac{1}{n-1}c_2\left(t-\frac{1}{2}\right)+O(1/2^n).$$

*Proof.* The residual after the first step is

$$f^*(s,t)-h^1(s,t) = c_2\left(t-\frac{\xi_{s1}}{\xi_{s0}+\xi_{s1}}\right).$$

Therefore, we get

$$h^2(s,t) = \frac{c_2}{\xi_{0t}+\xi_{1t}}\left(\left(t-\frac{\xi_{01}}{\xi_{00}+\xi_{01}}\right)\xi_{0t}+\left(t-\frac{\xi_{11}}{\xi_{10}+\xi_{11}}\right)\xi_{1t}\right),$$

which is equal to

$$-c_2\left(\frac{\xi_{00}\xi_{01}}{(\xi_{00}+\xi_{01})(\xi_{00}+\xi_{10})}+\frac{\xi_{10}\xi_{11}}{(\xi_{10}+\xi_{11})(\xi_{00}+\xi_{10})}\right)$$

for $t=0$ and to

$$c_2\left(\frac{\xi_{00}\xi_{01}}{(\xi_{00}+\xi_{01})(\xi_{01}+\xi_{11})}+\frac{\xi_{10}\xi_{11}}{(\xi_{10}+\xi_{11})(\xi_{01}+\xi_{11})}\right)$$

for $t=1$. The expected values of all four ratios are equal due to symmetries, and they are equal to $\frac{1}{4}\left(1-\frac{1}{n-1}\right)+O(2^{-n})$ according to Lemma 3 below. So, we obtain

$$\mathbb{E}(h^2(s,t) \mid A) = (2t-1)\frac{c_2}{2}\left(1-\frac{1}{n-1}\right)+O(2^{-n})$$

and

$$\mathbb{E}(h^1(s,t)+h^2(s,t) \mid A) = f^*(s,t) - c_2\frac{1}{n-1}\left(t-\frac{1}{2}\right)+O(2^{-n}).$$

□

**Lemma 3** *We have*

$$\mathbb{E}\left(\frac{\xi_{00}\xi_{01}}{(\xi_{00}+\xi_{01})(\xi_{01}+\xi_{11})} \mid A\right) = \frac{1}{4}\left(1-\frac{1}{n-1}\right)+O(2^{-n}).$$

*Proof.* First, linearity implies

$$\mathbb{E}\left(\frac{\xi_{00}\xi_{01}}{(\xi_{00}+\xi_{01})(\xi_{01}+\xi_{11})}\mid A\right)=\sum_{i,j}\mathbb{E}\left(\frac{\mathbb{1}_{\mathbf{x}_i=(0,0),\mathbf{x}_j=(0,1)}}{(\xi_{00}+\xi_{01})(\xi_{01}+\xi_{11})}\mid A\right).$$

Taking into account that all terms are equal, the expectation can be written as $\frac{n(n-1)}{4^2}a$, where

$$a=\mathbb{E}\left(\frac{1}{(\xi_{00}+\xi_{01})(\xi_{01}+\xi_{11})}\mid \mathbf{x}_1=(0,0),\mathbf{x}_2=(0,1),A\right).$$

A key observation is that $\xi_{00}+\xi_{01}$ and $\xi_{01}+\xi_{11}$ are two independent binomial variables: the former one is the number of $k$ such that $x_k^1=0$ and the latter one is the number of $k$ such that $x_k^2=1$. Moreover, they (and also their inverses) are also conditionally independent given that first two observations of the Bernoulli scheme are known ($\mathbf{x}_1=(0,0),\mathbf{x}_2=(0,1)$) and given $A$. This conditional independence implies that $a$ is the product of $\mathbb{E}\left(\frac{1}{\xi_{00}+\xi_{01}}\mid \mathbf{x}_1=(0,0),\mathbf{x}_2=(0,1),A\right)$ and $\mathbb{E}\left(\frac{1}{\xi_{01}+\xi_{11}}\mid \mathbf{x}_1=(0,0),\mathbf{x}_2=(0,1),A\right)$. The first factor equals $\frac{2}{n}\left(1-\frac{1}{n-1}+O(2^{-n})\right)$ according to Lemma 2. The second one is equal to $\mathbb{E}\left(\frac{1}{\xi_{01}+\xi_{11}}\mid \mathbf{x}_1=(0,0),\mathbf{x}_2=(0,1)\right)$ since $A$ does not bring any new information about the number of $k$ with $x_k^2=1$ given $\mathbf{x}_1=(0,0),\mathbf{x}_2=(0,1)$. So, according to Lemma 4 below, the second factor equals $\frac{2}{n-1}(1+O(2^{-n}))$. Finally, we obtain

$$\mathbb{E}\left(\frac{\xi_{00}\xi_{01}}{(\xi_{00}+\xi_{01})(\xi_{01}+\xi_{11})}\right)$$
$$=\frac{n(n-1)}{4^2}\frac{4}{n(n-1)}\left(1-\frac{1}{n-1}\right)+O(2^{-n})=\frac{1}{4}\left(1-\frac{1}{n-1}\right)+O(2^{-n}).$$

$\square$

**Lemma 4** $\mathbb{E}\left(\frac{1}{\xi_{01}+\xi_{11}}\mid \mathbf{x}_1=(0,0),\mathbf{x}_2=(0,1)\right)=\frac{2}{n-1}-\frac{1}{2^{n-2}(n-1)}.$

*Proof.* Similarly to the proof of Lemma 2, we have

$$\mathrm{P}(\xi_{01}+\xi_{11}=k\mid \mathbf{x}_1=(0,0),\mathbf{x}_2=(0,1))=\binom{n-2}{k-1}2^{-(n-2)}.$$

Therefore, we get

$$\mathbb{E}\left(\frac{1}{\xi_{01}+\xi_{11}}\mid \mathbf{x}_1=(0,0),\mathbf{x}_2=(0,1)\right)=\sum_{k=1}^{n-1}\frac{1}{k}\binom{n-2}{k-1}2^{-(n-2)}$$
$$=\frac{2^{-(n-2)}}{n-1}\sum_{k=1}^{n-1}\binom{n-1}{k}=\frac{2}{n-1}-\frac{1}{2^{n-2}(n-1)}.$$

$\square$

### A.2 Proof for independently sampled $\mathcal{D}_1$ and $\mathcal{D}_2$

Assume that we have an additional sample $\mathcal{D}_2=\{\mathbf{x}_{n+k}\}_{k=1..n}$ for building $h^2$. Now $A$ denotes the event that each leaf in $h^1$ contains at least one example from $\mathcal{D}_1$ and each leaf in $h^2$ contains at least one example from $\mathcal{D}_2$.

**Proposition 4** *If $h^2$ is built using dataset $\mathcal{D}_2$, then*

$$\mathbb{E}(h^1(s,t)+h^2(s,t)\mid A)=f^*(s,t).$$

*Proof.*

Let us denote by $\xi'_{st}$ the number of examples $\mathbf{x}_{n+k}$ that are equal to $(s,t)$, $k=1,\ldots,n$.

First, we need to derive the expectation $\mathbb{E}(h^2(s,t))$ of $h^2$ on a test example $\mathbf{x} = (s,t)$. Similarly to the proof of Proposition 3, we get

$$h^2(s,0) = -c_2 \left( \frac{\xi'_{00}\xi_{01}}{(\xi_{00} + \xi_{01})(\xi'_{00} + \xi'_{10})} + \frac{\xi'_{10}\xi_{11}}{(\xi_{10} + \xi_{11})(\xi'_{00} + \xi'_{10})} \right) \,,$$

$$h^2(s,1) = c_2 \left( \frac{\xi_{00}\xi'_{01}}{(\xi_{00} + \xi_{01})(\xi'_{01} + \xi'_{11})} + \frac{\xi_{10}\xi'_{11}}{(\xi_{10} + \xi_{11})(\xi'_{01} + \xi'_{11})} \right) \,.$$

Due to the symmetries, the expected values of all four fractions above are equal. Also, due to the independence of $\xi_{ij}$ and $\xi'_{kl}$, we have

$$\mathbb{E}\left( \frac{\xi'_{00}\xi_{01}}{(\xi_{00} + \xi_{01})(\xi'_{00} + \xi'_{10})} \mid A \right) = \mathbb{E}\left( \frac{\xi_{01}}{\xi_{00} + \xi_{01}} \mid A \right) \mathbb{E}\left( \frac{\xi'_{00}}{\xi'_{00} + \xi'_{10}} \mid A \right) = \frac{1}{4} \,.$$

Therefore, $\mathbb{E}(h^2(s,0) \mid A) = -\frac{c_2}{2}$ and $\mathbb{E}(h^2(s,1) \mid A) = \frac{c_2}{2}$.

Summing up, $\mathbb{E}(h^2(s,t) \mid A) = c_2 t - \frac{c_2}{2}$ and $\mathbb{E}(h^1(s,t) + h^2(s,t) \mid A) = c_1 s + c_2 t$. $\square$

## B   Formal description of CatBoost algorithm

In this section, we formally describe the CatBoost algorithm introduced in Section 5 of the main text. In Algorithm 1, we provide more information on particular details including the speeding up trick introduced in paragraph "Complexity" of the main text. The key step of the CatBoost algorithm is the procedure of building a tree described in detail in Function $BuildTree$. To obtain the formal description of the CatBoost algorithm without the speeding up trick, one should replace $\lceil \log_2 n \rceil$ by $n$ in line 6 of Algorithm 1 and use Algorithm 2 from the main text instead of Function $BuildTree$.

We use Function $GetLeaf(\mathbf{x}, T, \sigma_r)$ to describe how examples are matched to leaves $leaf_r(i)$. Given an example with features $\mathbf{x}$, we calculate ordered TS on the basis of the permutation $\sigma_r$ and then choose the leaf of tree $T$ corresponding to features $\mathbf{x}$ enriched by the obtained ordered TS. Using $ApplyMode$ instead of a permutation in function $GetLeaf$ in line 15 of Algorithm 1 means that we use TS calculated on the whole training data to apply the trained model on a new example.

---

**Algorithm 1:** CatBoost

**input**  : $\{(\mathbf{x}_i, y_i)\}_{i=1}^n$, $I$, $\alpha$, $L$, $s$, $Mode$

1   $\sigma_r \leftarrow$ random permutation of $[1, n]$ for $r = 0..s$;
2   $M_0(i) \leftarrow 0$ for $i = 1..n$;
3   **if** $Mode = Plain$ **then**
4     $\lfloor$ $M_r(i) \leftarrow 0$ for $r = 1..s$, $i : \sigma_r(i) \le 2^{j+1}$;
5   **if** $Mode = Ordered$ **then**
6     **for** $j \leftarrow 1$ **to** $\lceil \log_2 n \rceil$ **do**
7      $\lfloor$ $M_{r,j}(i) \leftarrow 0$ for $r = 1..s$, $i = 1..2^{j+1}$;

8   **for** $t \leftarrow 1$ **to** $I$ **do**
9     $T_t, \{M_r\}_{r=1}^s \leftarrow BuildTree(\{M_r\}_{r=1}^s, \{(\mathbf{x}_i, y_i)\}_{i=1}^n, \alpha, L, \{\sigma_i\}_{i=1}^s, Mode)$;
10     $leaf_0(i) \leftarrow GetLeaf(\mathbf{x}_i, T_t, \sigma_0)$ for $i = 1..n$;
11     $grad_0 \leftarrow CalcGradient(L, M_0, y)$;
12     **foreach** $leaf\ j$ in $T_t$ **do**
13      $\lfloor$ $b_j^t \leftarrow -\text{avg}(grad_0(i)$ for $i : leaf_0(i) = j)$;
14     $M_0(i) \leftarrow M_0(i) + \alpha b_{leaf_0(i)}^t$ for $i = 1..n$;

15   **return** $F(\mathbf{x}) = \sum_{t=1}^I \sum_j \alpha b_j^t \mathbb{1}_{\{GetLeaf(\mathbf{x}, T_t, ApplyMode)=j\}}$;

---

**Function** $BuildTree$

**input** : $M, \{(\mathbf{x}_i, y_i)\}_{i=1}^n, \alpha, L, \{\sigma_i\}_{i=1}^s, Mode$

1  $grad \leftarrow CalcGradient(L, M, y)$;
2  $r \leftarrow random(1, s)$;
3  **if** $Mode = Plain$ **then**
4  $\quad\lfloor\ G \leftarrow (grad_r(i) \text{ for } i = 1..n)$;
5  **if** $Mode = Ordered$ **then**
6  $\quad\lfloor\ G \leftarrow (grad_{r, \lfloor \log_2(\sigma_r(i)-1)\rfloor}(i) \text{ for } i = 1..n)$;
7  $T \leftarrow$ empty tree;
8  **foreach** *step of top-down procedure* **do**
9  $\quad$ **foreach** *candidate split c* **do**
10 $\qquad T_c \leftarrow$ add split $c$ to $T$;
11 $\qquad leaf_r(i) \leftarrow GetLeaf(\mathbf{x}_i, T_c, \sigma_r) \text{ for } i = 1..n$;
12 $\qquad$ **if** $Mode = Plain$ **then**
13 $\qquad\quad\lfloor\ \Delta(i) \leftarrow \text{avg}(grad_r(p) \text{ for } p : \ leaf_r(p) = leaf_r(i)) \ \text{ for } i = 1..n$;
14 $\qquad$ **if** $Mode = Ordered$ **then**
15 $\qquad\quad\lfloor\ \Delta(i) \leftarrow \text{avg}(grad_{r, \lfloor \log_2(\sigma_r(i)-1)\rfloor}(p) \text{ for } p : \ leaf_r(p) = leaf_r(i), \sigma_r(p) < \sigma_r(i)) \ \text{ for } i = 1..n$;
16 $\qquad\lfloor\ loss(T_c) \leftarrow \cos(\Delta, G)$
17 $\quad\lfloor\ T \leftarrow \arg\min_{T_c}(loss(T_c))$
18 $leaf_{r'}(i) \leftarrow GetLeaf(\mathbf{x}_i, T, \sigma_{r'}) \text{ for } r' = 1..s, i = 1..n$;
19 **if** $Mode = Plain$ **then**
20 $\quad\lfloor\ M_{r'}(i) \leftarrow M_{r'}(i) - \alpha \text{ avg}(grad_{r'}(p) \text{ for } p : \ leaf_{r'}(p) = leaf_{r'}(i)) \text{ for } r' = 1..s, i = 1..n$;
21 **if** $Mode = Ordered$ **then**
22 $\quad$ **for** $j \leftarrow 1$ **to** $\lceil \log_2 n \rceil$ **do**
23 $\qquad M_{r',j}(i) \leftarrow M_{r',j}(i) - \alpha \text{ avg}(grad_{r',j}(p) \text{ for } p : \ leaf_{r'}(p) = leaf_{r'}(i), \sigma_{r'}(p) \le 2^j) \text{ for } r' = 1..s, \ i : \sigma_{r'}(i) \le 2^{j+1}$;

24 **return** $T, M$

## C   Time complexity analysis

### C.1   Theoretical analysis

We present the computational complexity of different components of any of the two modes of CatBoost per one iteration in Table 1.

Table 1: Computational complexity.

| Procedure | CalcGradient | Build $T$ | Calc values $b_j^t$ | Update $M$ | Calc ordered TS |
|---|---|---|---|---|---|
| Complexity for iteration $t$ | $O(s \cdot n)$ | $O(|C| \cdot n)$ | $O(n)$ | $O(s \cdot n)$ | $O(N_{TS,t} \cdot n)$ |

We first prove these asymptotics for the Ordered mode. For this purpose, we estimate the number $N_{pred}$ of predictions $M_{r,j}(i)$ to be maintained:

$$N_{pred} = (s+1) \cdot \sum_{j=1}^{\lceil \log_2 n \rceil} 2^{j+1} < (s+1) \cdot 2^{\log_2 n + 3} = 8(s+1)n \,.$$

Then, obviously, the complexity of CalcGradient is $O(N_{pred}) = O(s \cdot n)$. The complexity of leaf values calculation is $O(n)$, since each example $i$ is included only in averaging operation in leaf $leaf_0(i)$.

Calculation of the ordered TS for one categorical feature can be performed sequentially in the order of the permutation by $n$ additive operations for calculation of $n$ partial sums and $n$ division operations.

Table 2: Comparison of running times on Epsilon

|  | time per tree |
|---|---|
| CatBoost Plain | **1.1 s** |
| CatBoost Ordered | 1.9 s |
| XGBoost | 3.9 s |
| LightGBM | **1.1 s** |

Thus, the overall complexity of the procedure is $O(N_{TS,t} \cdot n)$, where $N_{TS,t}$ is the number of TS which were not calculated on the previous iterations. Since the leaf values $\Delta(i)$ calculated in line 15 of Function $BuildTree$ can be considered as ordered TS, where gradients play the role of targets, the complexity of building a tree $T$ is $O(|C| \cdot n)$, where $C$ is the set of candidate splits to be considered at the given iteration. Finally, for updating the supporting models (lines 22-23 in Function $BuildTree$), we need to perform one averaging operation for each $j = 1, \dots, \lceil \log_2 n \rceil$, and each maintained gradient $grad_{r',j}(p)$ is included in one averaging operation. Thus, the number of operations is bounded by the number of the maintained gradients $grad_{r',j}(p)$, which is equal to $N_{pred} = O(s \cdot n)$.

To finish the proof, note that any component of the Plain mode is not less efficient than the same one of the Ordered mode but, at the same time, cannot be more efficient than corresponding asymptotics from Table 1.

### C.2 Empirical analysis

It is quite hard to compare different boosting libraries in terms of training speed. Every algorithm has a vast number of parameters which affect training speed, quality and model size in a non-obvious way. Different libraries have their unique quality/training speed trade-off's and they cannot be compared without domain knowledge (e.g., is $0.5\%$ of quality metric worth it to train a model 3-4 times slower?). Plus for each library it is possible to obtain almost the same quality with different ensemble sizes and parameters. As a result, one cannot compare libraries by time needed to obtain a certain level of quality. As a result, we could give only some insights of how fast our implementation could train a model of a fixed size. We use Epsilon dataset and we measure mean tree construction time one can achieve without using feature subsampling and/or bagging by CatBoost (both Ordered and Plain modes), XGBoost (we use histogram-based version, which is faster) and LightGBM. For XGBoost and CatBoost we use the default tree depth equal to 6, for LightGBM we set leaves count to 64 to have comparable results. We run all experiments on the same machine with Intel Xeon E3-12xx 2.6GHz, 16 cores, 64GB RAM and run all algorithms with 16 threads.

We set such learning rate that algorithms start to overfit approximately after constructing about 7000 trees and measure the average time to train ensembles of 8000 trees. Mean tree construction time is presented in Table 2. Note that CatBoost Plain and LightGBM are the fastest ones followed by Ordered mode, which is about 1.7 times slower, which is expected.

Finally, let us note that CatBoost has a highly efficient GPU implementation. The detailed description and comparison of the running times are beyond the scope of the current article, but these experiments can be found on the corresponding GitHub page.[1]

## D Experimental setup

### D.1 Description of the datasets

The datasets used in our experiments are described in Table 3.

Table 3: Description of the datasets.

| Dataset name | Instances | Features | Description |
|---|---|---|---|
| Adult[2] | 48842 | 15 | Prediction task is to determine whether a person makes over 50K a year. Extraction was done by Barry Becker from the 1994 Census database. A set of reasonably clean records was extracted using the following conditions: (AAGE>16) and (AGI>100) and (AFNLWGT>1) and (HRSWK>0) |
| Amazon[3] | 32769 | 10 | Data from the Kaggle Amazon Employee challenge. |
| Click Prediction[4] | 399482 | 12 | This data is derived from the 2012 KDD Cup. The data is subsampled to 1% of the original number of instances, downsampling the majority class (click=0) so that the target feature is reasonably balanced (5 to 1). The data is about advertisements shown alongside search results in a search engine, and whether or not people clicked on these ads. The task is to build the best possible model to predict whether a user will click on a given ad. |
| Epsilon[5] | 400000 | 2000 | PASCAL Challenge 2008. |
| KDD appetency[6] | 50000 | 231 | Small version of KDD 2009 Cup data. |
| KDD churn[7] | 50000 | 231 | Small version of KDD 2009 Cup data. |
| KDD Internet[8] | 10108 | 69 | Binarized version of the original dataset. The multi-class target feature is converted to a two-class nominal target feature by re-labeling the majority class as positive ('P') and all others as negative ('N'). Originally converted by Quan Sun. |
| KDD upselling[9] | 50000 | 231 | Small version of KDD 2009 Cup data. |
| Kick prediction[10] | 72983 | 36 | Data from "Don't Get Kicked!" Kaggle challenge. |

## D.2 Experimental settings

In our experiments, we evaluate different modifications of CatBoost and two popular gradient boosting libraries: LightGBM and XGBoost. All the code needed for reproducing our experiments is published on our GitHub[11].

**Train-test splits** Each dataset was randomly split into training set (80%) and test set (20%). We denote them as $D_{full\_train}$ and $D_{test}$.

We use 5-fold cross-validation to tune parameters of each model on the training set. Accordingly, $D_{full\_train}$ is randomly split into 5 equally sized parts $D_1, \ldots, D_5$ (sampling is stratified by classes). These parts are used to construct 5 training and validation sets: $D_i^{train} = \cup_{j \neq i} D_j$ and $D_i^{val} = D_i$ for $1 \leq i \leq 5$.

**Preprocessing** We applied the following steps to datasets with missing values:

---

[2]https://archive.ics.uci.edu/ml/datasets/Adult

[3]https://www.kaggle.com/c/amazon-employee-access-challenge

[4]http://www.kdd.org/kdd-cup/view/kdd-cup-2012-track-2

[5]https://www.csie.ntu.edu.tw/~cjlin/libsvmtools/datasets/binary.html

[6]http://www.kdd.org/kdd-cup/view/kdd-cup-2009/Data

[7]http://www.kdd.org/kdd-cup/view/kdd-cup-2009/Data

[8]https://kdd.ics.uci.edu/databases/internet_usage/internet_usage.html

[9]http://www.kdd.org/kdd-cup/view/kdd-cup-2009/Data

[10]https://www.kaggle.com/c/DontGetKicked

[11]https://github.com/catboost/benchmarks/tree/master/quality_benchmarks

- For categorical variables, missing values are replaced with a special value, i.e., we treat missing values as a special category;

- For numerical variables, missing values are replaced with zeros, and a binary dummy feature for each imputed feature is added.

For XGBoost, LightGBM and the raw setting of CatBoost (see Section G), we perform the following preprocessing of categorical features. For each pair of datasets $(D_i^{train}, D_i^{val})$, $i = 1, \ldots, 5$, and $(D_{full\_train}, D_{test})$, we preprocess the categorical features by calculating ordered TS (described in Section 3.2 of the main text) on the basis of a random permutation of the examples of the first (training) dataset. All the permutations are generated independently. The resulting values of TS are considered as numerical features by any algorithm to be evaluated.

**Parameter Tuning**  We tune all the key parameters of each algorithm by 50 steps of the sequential optimization algorithm Tree Parzen Estimator implemented in Hyperopt library[12] (mode *algo=tpe.suggest*) by minimizing logloss. Below is the list of the tuned parameters and their distributions the optimization algorithm started from:

XGBoost:

- 'eta': Log-uniform distribution $[e^{-7}, 1]$
- 'max_depth': Discrete uniform distribution $[2, 10]$
- 'subsample': Uniform $[0.5, 1]$
- 'colsample_bytree': Uniform $[0.5, 1]$
- 'colsample_bylevel': Uniform $[0.5, 1]$
- 'min_child_weight': Log-uniform distribution $[e^{-16}, e^5]$
- 'alpha': Mixed: $0.5 \cdot$ Degenerate at $0 + 0.5 \cdot$ Log-uniform distribution $[e^{-16}, e^2]$
- 'lambda': Mixed: $0.5 \cdot$ Degenerate at $0 + 0.5 \cdot$ Log-uniform distribution $[e^{-16}, e^2]$
- 'gamma': Mixed: $0.5 \cdot$ Degenerate at $0 + 0.5 \cdot$ Log-uniform distribution $[e^{-16}, e^2]$

LightGBM:

- 'learning_rate': Log-uniform distribution $[e^{-7}, 1]$
- 'num_leaves' : Discrete log-uniform distribution $[1, e^7]$
- 'feature_fraction': Uniform $[0.5, 1]$
- 'bagging_fraction': Uniform $[0.5, 1]$
- 'min_sum_hessian_in_leaf': Log-uniform distribution $[e^{-16}, e^5]$
- 'min_data_in_leaf': Discrete log-uniform distribution $[1, e^6]$
- 'lambda_l1': Mixed: $0.5 \cdot$ Degenerate at $0 + 0.5 \cdot$ Log-uniform distribution $[e^{-16}, e^2]$
- 'lambda_l2': Mixed: $0.5 \cdot$ Degenerate at $0 + 0.5 \cdot$ Log-uniform distribution $[e^{-16}, e^2]$

CatBoost:

- 'learning_rate': Log-uniform distribution $[e^{-7}, 1]$
- 'random_strength': Discrete uniform distribution over a set $\{1, 20\}$
- 'one_hot_max_size': Discrete uniform distribution over a set $\{0, 25\}$
- 'l2_leaf_reg': Log-uniform distribution $[1, 10]$
- 'bagging_temperature': Uniform $[0, 1]$
- 'gradient_iterations' : Discrete uniform distribution over a set $\{1, 10\}$

Next, having fixed all other parameters, we perform exhaustive search for the number of trees in the interval $[1, 5000]$. We collect logloss value for each training iteration from 1 to 5000 for each of the 5 folds. Then we choose the iteration with minimum logloss averaged over 5 folds.

For evaluation, each algorithm was run on the preprocessed training data $D_{full\_train}$ with the tuned parameters. The resulting model was evaluated on the preprocessed test set $D_{test}$.

**Versions of the libraries**

- catboost (0.3)
- xgboost (0.6)
- scikit-learn (0.18.1)
- scipy (0.19.0)
- pandas (0.19.2)
- numpy (1.12.1)
- lightgbm (0.1)
- hyperopt (0.0.2)
- h2o (3.10.4.6)
- R (3.3.3)

# E    Analysis of iterated bagging

Based on the out-of-bag estimation [1], Breiman proposed *iterated bagging* [2] which simultaneously constructs $K$ models $F_i$, $i = 1, \ldots, K$, associated with $K$ independently bootstrapped subsamples $\mathcal{D}_i$. At $t$-th step of the process, models $F_i^t$ are grown from their predecessors $F_i^{t-1}$ as follows. The current estimate $M_j^t$ at example $j$ is obtained as the average of the outputs of all models $F_k^{t-1}$ such that $j \notin \mathcal{D}_k$. The term $h_i^t$ is built as a predictor of the residuals $r_j^t := y_j - M_j^t$ (targets minus current estimates) on $\mathcal{D}_i$. Finally, the models are updated: $F_i^t := F_i^{t-1} + h_i^t$. Unfortunately, the residuals $r_j^t$ used in this procedure are not unshifted (in terms of Section 4.1 of the main text), or unbiased (in terms of *iterated bagging*), because each model $F_i^t$ depends on each observation $(\mathbf{x}_j, y_j)$ by construction. Indeed, although $h_k^t$ does not use $y_j$ directly, if $j \notin \mathcal{D}_k$, it still uses $M_{j'}^{t-1}$ for $j' \in \mathcal{D}_k$, which, in turn, can depend on $(\mathbf{x}_j, y_j)$.

Also note that computational complexity of this algorithm exceeds one of classic GBDT by factor of $K$.

# F    Ordered boosting with categorical features

In Sections 3.2 and 4.2 of the main text, we proposed to use some random permutations $\sigma_{cat}$ and $\sigma_{boost}$ of training examples for the TS calculation and for ordered boosting, respectively. Now, being combined in one algorithm, should these two permutations be somehow dependent? We argue that they should coincide. Otherwise, there exist examples $\mathbf{x}_i$ and $\mathbf{x}_j$ such that $\sigma_{boost}(i) < \sigma_{boost}(j)$ and $\sigma_{cat}(i) > \sigma_{cat}(j)$. Then, the model $M_{\sigma_{boost}(j)}$ is trained using TS features of, in particular, example $\mathbf{x}_i$, which are calculated using $y_j$. In general, it may shift the prediction $M_{\sigma_{boost}(j)}(\mathbf{x}_j)$. To avoid such a shift, we set $\sigma_{cat} = \sigma_{boost}$ in CatBoost. In the case of the ordered boosting (Algorithm 1 in the main text) with sliding window TS[13] it guarantees that the prediction $M_{\sigma(i)-1}(\mathbf{x}_i)$ is not shifted for $i = 1, \ldots, n$, since, first, the target $y_i$ was not used for training $M_{\sigma(i)-1}$ (neither for the TS calculation, nor for the gradient estimation) and, second, the distribution of TS $\hat{x}^i$ conditioned by the target value is the same for a training example and a test example with the same value of feature $x^i$.

# G  Experimental results

**Comparison with baselines**   In Section 6 of the main text we demonstrated that the strong setting of CatBoost, including ordered TS, Ordered mode and feature combinations, outperforms the baselines. Detailed experimental results of that comparison are presented in Table 4.

Table 4: Comparison with baselines: logloss / zero-one loss, relative increase is presented in the brackets.

|  | CatBoost | LightGBM | XGBoost |
|---|---|---|---|
| Adult | **0.2695 / 0.1267** | 0.2760 (+2.4%) / 0.1291 (+1.9%) | 0.2754 (+2.2%) / 0.1280 (+1.0%) |
| Amazon | **0.1394 / 0.0442** | 0.1636 (+17%) / 0.0533 (+21%) | 0.1633 (+17%) / 0.0532 (+21%) |
| Click | **0.3917 / 0.1561** | 0.3963 (+1.2%) / 0.1580 (+1.2%) | 0.3962 (+1.2%) / 0.1581 (+1.2%) |
| Epsilon | **0.2647 / 0.1086** | 0.2703 (+1.5%) / 0.114 (+4.1%) | 0.2993 (+11%) / 0.1276 (+12%) |
| Appetency | **0.0715 / 0.01768** | 0.0718 (+0.4%) / 0.01772 (+0.2%) | 0.0718 (+0.4%) / 0.01780 (+0.7%) |
| Churn | **0.2319 / 0.0719** | 0.2320 (+0.1%) / 0.0723 (+0.6%) | 0.2331 (+0.5%) / 0.0730 (+1.6%) |
| Internet | **0.2089 / 0.0937** | 0.2231 (+6.8%) / 0.1017 (+8.6%) | 0.2253 (+7.9%) / 0.1012 (+8.0%) |
| Upselling | **0.1662 / 0.0490** | 0.1668 (+0.3%) / 0.0491 (+0.1%) | 0.1663 (+0.04%) / 0.0492 (+0.3%) |
| Kick | **0.2855 / 0.0949** | 0.2957 (+3.5%) / 0.0991 (+4.4%) | 0.2946 (+3.2%) / 0.0988 (+4.1%) |

In this section, we empirically show that our implementation of GBDT provides state-of-the-art quality and thus is an appropriate basis for building CatBoost by adding different improving options including the above-mentioned ones. For this purpose, we compare with baselines a *raw setting* of CatBoost which is as close to classical GBDT [3] as possible. Namely, we use CatBoost in GPU mode with the following parameters: *−−boosting−type Plain  −−border−count 255  −−dev−bootstrap−type DiscreteUniform  −−gradient−iterations 1  −−random−strength 0  −−depth 6*. Besides, we tune the parameters *dev−sample−rate, learning−rate, l2−leaf−reg* instead of the parameters described in paragraph "Parameter tuning" of Section D.2 by 50 steps of the optimization algorithm. Further, for all the algorithms, all categorical features are transformed to ordered TS on the basis of a random permutation (the same for all algorithms) of training examples at the preprocessing step. The resulting TS are used as numerical features in the training process. Thus, no CatBoost options dealing with categorical features are used. As a result, the main difference of the raw setting of CatBoost compared with XGBoost and LightGBM is using oblivious trees as base predictors.

Table 5: Comparison with baselines: logloss / zero-one loss (relative increase for baselines).

|  | Raw setting of CatBoost | LightGBM | XGBoost |
|---|---|---|---|
| Adult | 0.2800 / 0.1288 | -1.4% / +0.2% | -1.7% / -0.6% |
| Amazon | 0.1631 / 0.0533 | +0.3% / 0% | +0.1% / -0.2% |
| Click | 0.3961 / 0.1581 | +0.1% / -0.1% | 0% / 0% |
| Appetency | 0.0724 / 0.0179 | -0.8% / -1.0% | -0.8% / -0.4% |
| Churn | 0.2316 / 0.0718 | +0.2% / +0.7% | +0.6% / +1.6% |
| Internet | 0.2223 / 0.0993 | +0.4% / +2.4% | +1.4% / +1.9% |
| Upselling | 0.1679 / 0.0493 | -0.7% / -0.4% | -1.0% / -0.2% |
| Kick | 0.2955 / 0.0993 | +0.1% / -0.4% | -0.3% / -0.2% |
| Average |  | -0.2% / +0.2% | -0.2% / +0.2% |

For the baselines, we take the same results as in Table 4. As we can see from Table 5, in average, the difference between all the algorithms is rather small: the raw setting of CatBoost outperforms the baselines in terms of zero-one loss by 0.2% while they are better in terms of logloss by 0.2%. Thus, taking into account that a GBDT model with oblivious trees can significantly speed up execution at testing time [4], our implementation of GBDT is very reasonable choice to build CatBoost on.

**Ordered and Plain modes**   In Section 6 of the main text we showed experimentally that Ordered mode of CatBoost significantly outperforms Plain mode in the strong setting of CatBoost, including ordered TS and feature combinations. In this section, we verify that this advantage is not caused by interaction with these and other specific CatBoost options. For this purpose, we compare Ordered and Plain modes in the raw setting of CatBoost described in the previous paragraph.

In Table 6, we present relative results w.r.t. Plain mode for two modifications of Ordered mode. The first one uses one random permutation $\sigma_{boost}$ for Ordered mode generated independently from the permutation $\sigma_{cat}$ used for ordered TS. Clearly, discrepancy between the two permutations provides target leakage, which should be avoided. However, even in this setting Ordered mode considerably outperforms Plain one by 0.5% in terms of logloss and by 0.2% in terms of zero-one loss in average. Thus, advantage of Ordered mode remains strong in the raw setting of CatBoost.

Table 6: Ordered vs Plain modes in raw setting: change of logloss / zero-one loss relative to Plain mode.

|  | Ordered, $\sigma_{boost}$ independent of $\sigma_{cat}$ | Ordered, $\sigma_{boost} = \sigma_{cat}$ |
| --- | --- | --- |
| Adult | -1.1% / +0.2% | -2.1% / -1.2% |
| Amazon | +0.9% / +0.9% | +0.8% / -2.2% |
| Click | 0% / 0% | 0.1% / 0% |
| Appetency | -0.2% / 0.2% | -0.5% / -0.3% |
| Churn | +0.2% / -0.1% | +0.3% / +0.4% |
| Internet | -3.5% / -3.2% | -2.8% / -3.5% |
| Upselling | -0.4% / +0.3% | -0.3% / -0.1% |
| Kick | -0.2% / -0.1% | -0.2% / -0.3% |
| Average | -0.5% / -0.2% | -0.6% / -0.9% |

In the second modification, we set $\sigma_{boost} = \sigma_{cat}$, which remarkably improves both metrics: the relative difference with Plain becomes (in average) 0.6% for logloss and 0.9% for zero-one loss. This result empirically confirms the importance of the correspondence between permutations $\sigma_{boost}$ and $\sigma_{cat}$, which was theoretically motivated in Section F.

**Feature combinations**  To demonstrate the effect of feature combinations, in Figure 1 we present the relative change in logloss for different numbers $c_{max}$ of features allowed to be combined (compared to $c_{max} = 1$, where combinations are absent). In average, changing $c_{max}$ from 1 to 2 provides an outstanding improvement of 1.86% (reaching 11.3%), changing from 1 to 3 yields 2.04%, and further increase of $c_{max}$ does not influences the performance significantly.

Figure 1: Relative change in logloss for a given allowed complexity compared to the absence of feature combinations.

**Number of permutations**  The effect of the number $s$ of permutations on the performance of CatBoost is presented in Figure 2. In average, increasing $s$ slightly decreases logloss, e.g., by 0.19% for $s = 3$ and by 0.38% for $s = 9$ compared to $s = 1$.

## Footnotes

[1]`https://github.com/catboost/benchmarks/tree/master/gpu_training`

[12]https://github.com/hyperopt/hyperopt

[13]Ordered TS calculated on the basis of a fixed number of preceding examples (both for training and test examples).