[Reviews · NeurIPS 2018]

Reviewer 1



UPDATE AFTER AUTHORS' RESPONSE Regarding "using one tree structure", I think I understand now, and I think the current wording is confusing. Both the manuscript and the response made me think that the *same* tree splits (internal nodes) are used for all of the boosting iterations. But looking at the argmin line in Algorithm 2, I think the intent is to say "the same feature is used to split all internal nodes at a given level of a tree" (aka, oblivious tree). If that is not right, then I am still confused. Regarding one random permutation, please update text to be more clear. SUMMARY The paper identifies two potential sources of overfitting, one related to how high cardinality categorical features are encoded, and the other due to reuse of labeled examples across gradient boosting iterations. A new boosting variation, CatBoost, addresses both problems by treating the examples as an ordered sequence that is accessed in an online or prequential fashion (ala A.P. Dawid's work). The algorithm builds a nested sequence of models that are indexed against the sequence of labeled examples. Model i is allowed to use all the labeled examples seen before the i'th point in the example sequence, but none afterwards. This helps avoid estimation bias from reusing target labels (supported by theoretical analysis). Empirically, CatBoost is more accurate than popular boosting implementations (LightGBM and XGBoost) with comparable or faster training time. The impact from different components of the algorithm are measured empirically. REVIEW This is a very good paper. It makes several valuable contributions: * Help clarify a source of overfitting in boosting. This should help others continue research into the problem. * Propose a new, interesting way to avoid target leakage when representing categorical features with target statistics. The risk of target leakage is well-known, and the paper shows that the new approach leads to better accuracy than the simple approaches people are currently using. * Propose ordered boosting to solve the overfitting problem, and describe an efficient implementation of it. * Solid empirical study of the new algorithm. The main weakness is not including baselines that address the overfitting in boosting with heuristics. Ordered boosting is non-trivial, and it would be good to know how far simpler (heuristic) fixes go towards mitigating the problem. Overall, I think this paper will spur new research. As I read it, I easily came up with variations and alternatives that I wanted to see tried and compared. DETAILED COMMENTS The paper is already full of content, so the ideas for additional comparisons are really suggestions to consider. * For both model estimations, why start at example 1? Why not start at an example that is 1% of the way into the training data, to help reduce the risk of high variance estimates for early examples? * The best alternative I've seen for fixing TS leakage, while reusing the data sample, uses tools from differential privacy [1, 2]. How does this compare to Ordered TS? * Does importance-sampled voting [3] have the same target leakage problem as gradient boosting? This algorithm has a similar property of only using part of the sequence of examples for a given model. (I was very impressed by this algorithm when I used it; beat random forests hands down for our situation.) * How does ordered boosting compare to the subsampling trick mentioned in l. 150? * Yes, fixes that involve bagging (e.g., BagBoo [4]) add computational time, but so does having multiple permuted sequences. Seems worth a (future?) comparison. * Why not consider multiple permutations, and for each, split into required data subsets to avoid or mitigate leakage? Seems like it would have the same computational cost as ordered boosting. * Recommend checking out the Wilcoxon signed rank test for testing if two algorithms are significantly different over a range of data sets. See [6]. * l. 61: "A categorical feature..." * l. 73: "for each categorical *value*" ? * l. 97: For clarity, consider explaining a bit more how novel values in the test set are handled. * The approach here reminds me a bit of Dawid's prequential analysis, e.g., [5]. Could be worth checking those old papers to see if there is a useful connection. * l. 129: "we reveal" => "we describe" ? * l. 131: "called ordered boosting" * l. 135-137: The "shift" terminology seems less understandable than talking about biased estimates. * l. 174: "remind" => "recall" ? * l. 203-204: "using one tree structure"; do you mean shared \sigma? * Algorithm 1: only one random permutation? * l. 237: Don't really understand what is meant by right hand side of equality. What is 2^j subscript denoting? * l. 257: "tunning" => "tuning" * l. 268: ", what is expected." This reads awkwardly. * l. 311: This reference is incomplete. REFERENCES [1] https://www.slideshare.net/SessionsEvents/misha-bilenko-principal-researcher-microsoft [2] https://www.youtube.com/watch?v=7sZeTxIrnxs [3] Breiman (1999). Pasting small votes for classification in large databases and on-line. Machine Learning 36(1):85--103. [4] Pavlov et al. (2010). BagBoo: A scalable hybrid bagging-the-boosting model. In CIKM. [5] Dawid (1984). Present position and potential developments: Some personal views: Statistical Theory: The Prequential Approach. Journal of the Royal Stastical Society, Series A, 147(2). [6] Demsar (2006). Statistical comparisons of classifiers over multiple data sets. Journal of Machine Learning Research, 7:1--30.

Reviewer 2



The paper identifies and fixes a particular type of target leakage when processing categorical features for boosted decision trees. The authors fix this by imposing a (random) ordering of the samples and they use the same principle for building multiple trees. The paper shows strong empirical results against well known baselines. It is however hard to attribute the observed improvements to what the paper mostly talks about (i.e. the ordering principle). In the appendix for example, we see that a lot of lift comes from feature combinations. There are also other design decisions (e.g. use of oblivious trees) and I suspect the authors know how to tune the hyperparameters of their algorithm better than for the baselines. What would greatly strengthen the paper would be to show what would happen to XGBoost/LightGBM if the ordering principle was applied in their codebase. Otherwise we might be just observing an interesting interaction between ordering and some other design decisions in CatBoost. Finally one part that was not clear to me is why isn't there a new permutation in each iteration but instead only s of them.

Reviewer 3



Summary: This paper presents a set of novel tricks for gradient boosting toolkit called CatBoost. The main focus are to address two types of existing biases for (1) numerical values (calles TS, target statistics) that well summarize the categorical features (with high cardinality, in particular), and (2) gradient values of the current models required for each step of gradient boosting. For (1), one-hot encodings of categorical features could generate very sparse and high-dimensional features that cause computational difficulties, and surrogating by single TS values is a widely used trick to address it. But this TS for a categorical feature is computed using the target value of the same data, and thus results in a bias by some information leakage. Similarly for (2), the gradient values are computed using the same data that the current model is trained on. These two biases are addressed by random ordering and use the prefix of the sequences to balance the computational cost and the out-of-bag data availability. For (2), all random subsets of sample 1 to i (i for 1 to the sample size n) are kept and used in the algorithms. The experimental evaluations showed the superior performance against the state-of-the-art methods such as XGBoost and LightGBM. Strengths: - These reported biases were neither recognized nor previously addressed, and the tricks presented are very nicely simple in principle, and work well by elaborate implementation efforts. - The experimental performance are great compared to the state of the art methods such as XGBoost and LightGBM using the quite practical and very large-scale benchmark datasets. - The implementation would be very helpful for all practitioners in practice. Weakness: - Two types of methods are mixed into a single package (CatBoost) and evaluation experiments, and the contribution of each trick would be a bit unclear. In particular, it would be unclear whether CatBoost is basically for categorical data or it would also work with the numerical data only. - The bias under discussion is basically the ones occurred at each step, and their impact to the total ensemble is unclear. For example, randomization as seen in Friedman's stochastic gradient boosting can work for debiasing/stabilizing this type of overfitting biases. - The examples of Theorem 1 and the biases of TS are too specific and it is not convincing how these statement can be practical issues in general. Comment: - The main unclear point to me is whether CatBoost is mainly for categorical features or not. If the section 3 and 4 are independent, then it would be informative to separately evaluate the contribution of each trick. - Another unclear point is the paper presents specific examples of biases of target statistics (section 3.2) and prediction shift of gradient values (Theorem 1), and we can know that the bias can happen, but on the other hand, we are not sure how general these situations are. - One important thing I'm also interested in is that the latter bias 'prediction shift' is caused at each step, and its effect on the entire ensemble is not clear. For example, I guess the effect of the presented 'ordered boosting' could be related to Friedman's stochastic gradient boosting cited as [13]. This simple trick is just apply bagging to each gradient-computing step of gradient boosting, which would randomly perturb the exact computation of gradient. Each step would be just randomly biased, but the entire ensemble would be expected to be stabilized as a whole. Both XGBoost and LightGBM have this stochastic/bagging option, we can use it when we need it. Comment After Author Response: Thank you for the response. I appreciate the great engineering effort to realize a nice & high-performance implementations of CatBoost. But I'm still not sure that how 'ordering boosting', one of two main ideas of the paper, gives the performance improvement in general. As I mentioned in the previous comment, the bias occurs at each base learner h_t. But it is unclear that how this affects the entire ensemble F_t that we actually use. Since each h_t is a "weak" learner anyway, any small biases can be corrected to some extent through the entire boosting process. I couldn't find any comments for this point in the response. I understand the nice empirical results of Tab. 3 (Ordered vs. Plain gradient values) and Tab. 4 (Ordered TS vs. alternative TS methods). But I'm still unsure whether this improvement comes only from the 'ordering' ideas to address two types of target leakages. Because the comparing models have many different hyper parameters and (some of?) these are tuned by Hyperopt, so the improvement can come not only from addressing the two types of leakage. For example, it would be nice to have something like the following comparisons o focus only on two ideas of ordered TS and ordered boosting in addition: 1) Hyperopt-best-tuned comparisons of CatBoost (plain) vs LightGBM vs XGboost (to make sure no advantages exists for CatBoost (plain) ) 2) Hyperopt-best-tuned comparisons of CatBoost without column sampling + row sampling vs LightGBM/XGBoost without column sampling + row sampling 3) Hyperopt-best-tuned comparisons of CatBoost(plain) + ordered TS without ordered boosting vs CatBoost(plain) (any other randomization options, column sampling and row sampling, should be off) 4) Hyperopt-best-tuned comparisons of CatBoost(plain) + ordered boosting without ordered TS vs CatBoost(plain) (any other randomization options, column sampling and row sampling, should be off)